# The Selection of Gamma-Ray Irradiated Higher Yield Rice Mutants by Directed Evolution Method

**DOI:** 10.3390/plants9081004

**Published:** 2020-08-07

**Authors:** Hiroshi Kato, Feng Li, Akemi Shimizu

**Affiliations:** 1Genetic Resources Center, National Agriculture and Food Research Organization (NARO), 2-1-2 Kannondai, Tsukuba, Ibaraki 305-8602, Japan; 2Institute of Crop Science, National Agriculture and Food Research Organization (NARO), 2-1-2 Kannondai, Tsukuba, Ibaraki 305-8518, Japan; rihoumail@affrc.go.jp (F.L.); ami@affrc.go.jp (A.S.)

**Keywords:** higher yield, rice, mutant, gamma-ray irradiation, physiological trait, directed evolution

## Abstract

We have succeeded in selecting four higher yield mutants from five gamma-ray irradiated high-yielding Japanese rice varieties using a novel approach. A total of 464 M_2_ plants which had heavier total panicle weights per plant were first selected from 9801 irradiated M_2_ plants. Their higher yields were confirmed by yield trials conducted for three years with a six to ten-pairwise replicated plot design. FukuhibikiH6 and FukuhibikiH8 were selected from an irradiated high-yielding variety Fukuhibiki and showed 1.2% to 22.5% higher yield than their original significantly. YamadawaraH3 was selected from an irradiated high-yielding variety Yamadawara and its yield advantages were 2.7% to 3.9%. However, there was no difference in the genotypes of the 96 SNP (single nucleotide polymorphism) markers between the higher yield mutants and their respective original varieties. The differences in the measured phenotypical traits between each mutant and its original variety were not constant and the actual differences were marginal. Therefore, the higher yields of the selected mutants were likely to have been caused by physiological traits rather than phenotypical traits. The selection method used in this study is an application of the directed evolution method which has long been commonly used in the substantial improvements of microorganisms and their proteins.

## 1. Introduction

Yield is the most important trait of crops and it primarily determines the amount of crop production in a unit area or a nation [1]. Rice is one of the most important cereal crops in the world. A considerable number of studies have been conducted around the world on the breeding of high yield crops, including rice [2]. There are two important points about high yield studies. Firstly, a high yield study requires the determination of crop yield under the ideal cultivating conditions for the relevant crop. The yield under the ideal conditions is referred to as the intrinsic yield and its estimation is imperative in any high yield studies. Each yield trial is affected and often spoiled by its environmental factors and thus the intrinsic yield is masked by those factors. Unfortunately, conventional high yield rice studies have failed to take this important concept into account. The method in this study was designed to incorporate this point. Secondly, if we can dissect the genetic architecture of high yield, we can apply it to genomic selection, genetic modifications, or genome editing to pyramid multiple useful genes for obtaining higher yields. This is what we are working on and we look forward to providing further information on this in our future report.

Recently, quite a large number of reports about gene isolation and the identification of yield-related traits in rice have been published. Ashikari et al. and other researchers studied *Gn1a*, which is a gene for cytokinin oxidase/dehydrogenase. It increases the number of reproductive organs, resulting in the enhancement of grain productivity [3,4,5]. This *Gn1a* came from a high-yielding *indica* variety Habataki and was introduced into a Japanese low yield variety, Koshihikari. In these studies, they identified a high-yielding gene by crossing high yield and low yield varieties. The information on this gene is not so valuable for rice breeders since *Gn1a* had already been utilized among rice breeders before their studies and the gene is not effective to increase the yield of high-yielding variety Habataki.

Song et al. and other researchers studied the gene controlling grain size [6,7,8,9]. *GW2* is a RING-type protein with E3 ubiquitin ligase activity. The loss of *GW2* function increased cell numbers, resulting in larger spikelets, hulls, grain width, and higher yield. However, they used only 25 plants for yield determination. They also mentioned the necessity for further yield trials in plots with randomized blocks in paddies. Ujiie et al. studied *GS3*, also a gene for rice grain size, which did not show yield increase compared with both of the parental varieties [10]. *TGW6*, which is a gene for grain weight, increases the yield of Nipponbare, which was an old variety and not considered to be a high-yielding variety [11]. Xu et al. reviewed the dense and erect panicle 1 (*DEP1*) gene and studied some other panicle forms [12,13,14,15,16]. The yield advantage of *DEP1* was reported in comparison to those of Liaojian5 and Toyonishiki, which were old and not considered to be high-yielding varieties [17]. There are no clear-cut reports on *DEP1* causing higher yield than existing high-yielding varieties. None of the above studies are of proven practical value.

*NAL1*, which is identical to *GPS* (green for photosynthesis), was identified between the progeny of the cross combination of Takanari and Koshihikari [18,19]. The introduction of Takanari’s *GPS* into Koshihikari increased the photosynthesis rate of Koshihikari but not the yield. Fujita et al. also found *NAL1* from the new plant type cultivar YP9 that was backcrossed with IR64 as a recurrent parent [20,21]. They developed IRRI146-*SPIKE*, which showed 24% to 28% higher yield than its original high-yielding cultivar IR64. Murai et al. confirmed the higher yield of *Ur1* introduced lines [22]. *Ur1* (Undulate rachis-1) is an incompletely dominant rice gene found on chromosome 6. This is a classical phenotypic marker of rice, although the gene function has not yet been identified. Until now, *NAL1* from YP9 is the only rice gene whose function has been identified and showed higher yield than existing high-yielding varieties.

Li et.al. reported that they obtained higher yield rice mutants without phenotypic differences by using X-ray irradiation around sixty years ago [23]. However, there have been no further studies on this and there are no existing higher yield rice mutants available for analysis. In this study, we tried to acquire new intrinsic higher yield rice mutants from the offspring of gamma-ray irradiated high-yielding varieties. By gamma-ray irradiation, the number of mutated genes among the whole genome sequence is limited, so we can easily acquire practical mutants [24]. We could isolate the relevant rice gene for non-shattering with bulked whole-genome sequencing [25]. Having succeeded in isolating the relevant candidate genes for higher yield from the obtained higher yield mutants, our next target is to clarify the genes and identify their functions.

## 2. Results

### 2.1. Screening of the Candidate Mutant M_2_ Plants and M_3_ Lines

The total number of collected M_2_ plants was 9801 from five irradiated M_2_ varieties. Among them, 1600 were Fukuhibiki M_2_ and 55 M_2_ plants were selected from them according to the total panicle weight per plant. The plant yields of these selected M_2_ plants are shown in Appendix A. In our selection, we considered their positional advantage over other plants and their position in the field so that the effect of non-uniform soil fertility could be eliminated. For the other 4 M_2_ mutants, the number of harvested M_2_ plants were 3001, 3000, 2000, and 200 and the number of M_2_ plants selected according to their yield were 155, 132, 106, and 16 for Oonari, Yamadawara, Mochidawara, and Akidawara, respectively. Including Fukuhibiki M_2_, the total number of selected M_2_ plants and established M_3_ lines was 464.

Ten M_3_ plants that originated from one selected M_2_ plant were transplanted in 2015. We planted one of the original varieties as a comparison and it was followed by its nine mutant lines (Appendix A). This pattern was repeated and ended with the original variety. Seven plants in an M_3_ line of Fukuhibiki, excluding border plants, were harvested for yield evaluation. All the panicles of each plant were cut at the neck and collected into individual paper bags. The panicle weight of each plant was measured as the individual plant yield. We observed and recorded the heading dates, plant heights, leaf color, sterilities, and tiller formation of each line. Segregated lines were eliminated as candidates for the yield trials in the next step. In Appendix A, “Line number 325” was selected and a few plants were used for further evaluation. For establishing M_4_ lines from M_3_ lines, we selected the highest yield “Plant 6, 64.4 g” in “Line number 325”. We selected a total of 12 higher yield mutant candidates according to the yield performances of M_3_ lines in 2016.

### 2.2. Yield Evaluation of M_4_ Mutant Lines in 2016

Twelve higher yield candidates were tested in the yield trials in 2016. Table 1 and Appendix A show the performances of mutants with standard fertilizer application and heavy fertilizer application, respectively. The heading date of FukuhibikiH6 was the same as that of the original variety Fukuhibiki, while FukuhibikiH7 was one day earlier in Table 1. In addition to grain yield, brown rice weight is also an important target trait. Fukuhibiki and its mutants showed nearly 6 t/ha brown rice yield and this was higher than the average Japanese brown rice yield. Other varietal groups also showed higher yield. Lodging degrees of all the lines and varieties were limited. Major diseases and other causes of major losses were not observed. The conditions under which these yield evaluations were conducted were considered to be good. The brown rice yield of FukuhibikiH6 was higher than that of Fukuhibiki, but FukuhibikiH7 was lower than its original. The highest yield advantage among the 12 mutants over their respective originals was 7.5% this year and this was observed in the grain weight of YamadawaraH3 (Table 1). No significant differences were observed between any mutant and its original in two replications of this experimental design. Figure 1 showed OonariH1 and OonariH2. All the observed appearances of the mutants were quite similar to their original varieties.

### 2.3. Yield Evaluation of M_5_ Mutant Lines in 2017

We selected and evaluated five mutants, namely FukuhibikiH6, OonariH2, YamadawaraH2, YamadawaraH3, and MochidawaraH1, in 2017 according to the results of the previous yield trials conducted in 2016. In this yield trial, a ten-pairwise experimental design was used for the clarification of the difference between mutants and their original varieties. The blocks were arranged in numerical order (Appendix A). There were significant grain yield increases between three mutants and their original varieties. Appendix A shows the statistical procedure of grain yield of FukuhibikiH6 and Fukuhibiki in 2017. Other statistical significances were calculated in the same way. They were the pairs of FukuhibikiH6 and Fukuhibiki (7.98 and 7.59 t/ha, 5.3% higher), YamadawaraH3 and Yamadawara (9.71 and 9.36 t/ha, 3.7% higher), and MochidawaraH1 and Mochidawara (10.40 and 9.92 t/ha, 4.8% higher). By contrast, there was a significant grain yield decrease between OonariH2 and Oonari (10.44 and 10.84 t/ha, 3.5% lower). Appendix A shows the grain yield of FukuhibikiH6 and Fukuhibiki in each block. The fluctuations in the yields of both FukuhibikiH6 and its original followed the same undulating pattern over the blocks, reflecting the fact that any slight fluctuation in the fertility of the soil over the block had had the same degree of effect on both of them.

### 2.4. Evaluations of Yield and Yield-Related Traits of M_5_, M_6_, and M_7_ Mutant Lines in 2017, 2018, and 2019

Data on grain yield and yield-related traits of three mutants in 2017 are shown in Table 2. Data on yields and yield-related traits of four mutants in 2018 and 2019 are shown in Table 3 and Table 4.

FukuhibikiH6 and FukuhibikiH8 were selected from irradiated high yield variety Fukuhibiki and constantly showed a significantly higher yield than their original variety for three consecutive years. The grain yields of FukuhibikiH6 were 6.25 to 9.89 t/ha and the yield advantages were 1.2% to 8.5% over the original variety Fukuhibiki. The grain yields of FukuhibikiH8 were 7.23 to 10.18 t/ha and the yield advantages were 4.1% to 22.5% higher, too. The culm lengths of FukuhibikiH6 and FukuhibikiH8 were longer than their original in 2018 and 2019. The largest difference in the culm length between FukuhibikiH8 and its original was 6.6 cm in 2019 and this difference was the most obvious one among observable traits in the fields for all three years. One thousand grain weight of FukuhibikiH6 was heavier than its original in 2017 and 2019. The number of grains per panicle of FukuhibikiH6 and FukuhibikiH8 was larger than their original in 2019. The panicle length of FukuhibikiH6 was 0.5 cm and 0.9 cm longer in 2017 and 2019 with standard fertilizer application but 0.4 cm shorter with heavy fertilizer application in 2019. The lodging degrees of both mutants were worse than their original in 2019. The seed set percentages and the number of grains per panicle of FukuhibikiH8 were higher than its original in 2019. FukuhibikiH6 also showed similar tendencies in 2019. The 1000 thousand grain weights of Fukuhibiki and its mutants were lower in the heavy fertilizer application field than in the standard one in 2019. According to another data survey of the yield of Fukuhibiki [26], such lower 1000 grain weights in the fields with heavy fertilizer application were also observed in the Tohoku region. However, this was not statistically significant.

The grain yields of YamadawaraH3 were 9.71 and 10.18 t/ha in 2017 and 2018, and they were significantly higher than its original. Higher yield was also observed in 2019, though not statistically significant. Its panicle lengths were over 0.3 cm longer for all three years and culm lengths were also longer in 2017 and 2019. Its 1000 grain weight was higher in 2019 and its heading date was 3 days earlier than its original in 2018. Its lodging degree was worse than its original in 2019.

The grain yield of MochidawaraH1 was higher than its original in 2017, but the differences were not statistically significant in 2018 and 2019. Its seed set percentage was high in 2017. Its number of panicles and panicle length were smaller and shorter than its original in 2018 and 2019. Both the mutant and the original variety were lodged severely in 2019 due to typhoons during their maturation since they were late maturity lines and varieties. Partly because of the effect of the typhoon, the evaluation of MochidawaraH1 was not conclusive.

### 2.5. DNA Marker Analysis between Higher Yield Mutants and Their Original Varieties

To confirm that these mutants are not derived from an outcrossed line, we performed genotyping analysis using the 96 SNP (single nucleotide polymorphism) markers covering the whole genome. The results of SNP genotyping are shown in Figure 2. There were no differences between the three higher yield mutants and their irradiated original varieties. Only one difference was observed between a different stock Mochidawara and the original irradiated Mochidawara. These results clearly indicate that the materials in this experiment were genuine mutants from their original varieties. They were not cross-contaminations by unexpected cross-hybridization or mishandling seeds. These results also matched with our previous study that, by gamma-ray irradiation, found that the number of mutated genes among the whole genome sequence is limited [24].

Similarities between the SNP patterns of Fukuhibiki and Yamadawara were observed on chromosome 4, 7, and 12, and similarity of Yamadawara and Mochidawara on chromosome 6 was also observed. Fukuhibiki is a *japonica* variety and Mochidawara is an *indica* variety. Yamadawara is a crossbreed of a *japonica* line and another *japonica* and *indica* crossbreed. According to the SNP analysis, the pattern of Yamadawara matches with the information about parentages.

## 3. Discussion

### 3.1. Rice Higher Yield Mutants and Their Gene Identifications

Through the higher yield selection from 9801 M_2_ plants and their lines in the following generations, we could select four intrinsic higher yield mutants which were statistically confirmed by multi-replicated and multi-year yield trails. Among the four mutants, the higher yields of FukuhibikiH6, FukuhibikiH8, and YamadawaraH3 were expressed constantly, but the higher yield of MochidawaraH1 was observed only in 2017, while its yields were similar to its original in 2018 and 2019. We may conclude that even a line possessing the higher yield trait may not always express its higher yield in the field, depending on the specific environmental conditions of the year. Theoretically speaking, MochidawaraH1 is still potentially useful because its yields were higher or at least the same as its original, not lower. However, the isolation of its relevant gene or genes and further functional study will be difficult because of its low frequency expression.

Recently, an approach combining bulked segregant analysis with whole-genome resequencing has dramatically accelerated the process of identifying candidate mutations. By using gamma-ray irradiation, we managed to identify relevant gene candidates on chromosome 3 for higher yield rice mutant FukuhibikiH8 and on chromosome 5 for YamadawaraH3 [27]. Further confirmation and identification of isolated genes are necessary, but relevant gene candidates have already been isolated in two of the three high frequency expression mutant lines, FukuhibikiH6, FukuhibikiH8, and YamadawaraH3. So far, few genes have been made available for genome editing targeting higher yield. The genes found here will be useful for genome editing for developing much higher yield crops. These findings can be also be used for conventional cross-hybridization breeding in rice.

### 3.2. The Comparison between Higher Yield Mutants Obtained in This Report and Previously Reported High-Yielding Materials

Under the present circumstances, knowledge about higher yield genes that can contribute to higher yield breeding is limited. Higher yield gene explorations using high and low yield cross-combinations have not been successful since it is difficult to overcome the high-yielding parents by using such combinations. Fujita et al. found *NAL1*/*SPIKE* from the new plant type cultivar YP9 (Daringan, IR68522-10-2-2), which showed 24% to 28% higher yield than its original high-yielding cultivar, IR64 [20,21]. In these reports, higher yield traits of YP9 were introduced into IR64. H. Kato, one of the authors of this report, contributed to the identification of *NAL1* by selecting the line which harbored the *SPIKE*/*NAL1* gene from YP9 [28]. In addition to *NAL1*, we have succeeded in finding higher yield genes in mutants in this report. One of the common factors of *NAL1* and the higher yield mutants that we have found is that the higher yield sources are introduced from exotic materials. In this report, they are the mutants from gamma-ray irradiation, and in the case of *NAL1,* it was YP9. *NAL1* gene identification took 16 years from the time that IR64 crossed with YP9 in 1997 to gene identification through marker aided selection and analysis of transformants in 2013. In this report, it took only 6 years from the irradiation of high-yielding varieties in 2013 to the isolation of higher yield candidate genes in 2019 [27]. This is a much more rapid method compared to the conventional marker aided selection.

Historically crop mutation breeding has been conducted for clearly distinguishable traits [29]. The selection of higher yield mutants without consideration of yield-related phenotypical traits has been extremely limited. Li et al. reported that they obtained higher yield rice mutants by using X-ray irradiation around sixty years ago [25]. The higher yield mutants that they selected did not show apparent differences from those of their original varieties. However, there was no description of the screening methods and there were no available technologies at that time for the isolation of the relevant gene. At this moment, there is no way to confirm their studies. Practically speaking, this is the first report on the selection of higher yield crop mutants that does not depend on the selection of yield-related phenotypical traits.

### 3.3. The Method in This Report Is Considered to Be an Application of the “Directed Evolution Method”

The selection method here is considered to be an application of the directed evolution method, a process that mimics Darwinian selection on a laboratory scale, and has enabled the evolution of virtually any protein, pathway, network, or even an entire organism [30,31]. Directed evolution can be defined as an iterative two-step process involving, firstly, the generation of a library of variants of a biological entity of interest and, secondly, the screening of this library to identify those mutants that exhibit better properties like higher activity. These studies have been conducted in the laboratory on synthetic biology, like proteins or their genomes. In plants, there were proposals for the combination of the directed evolution method and CRISPR/Cas9 genome editing technology for a new green revolution [32,33]. However, technologies including CRISPR/Cas9 have some problems that need to be solved [34]. Nevertheless, crop improvement using CRISPR-Cas9 usually uses information of known genes, of which, in fact, the number is very limited.

Conventionally, breeders conduct yield trials only in advanced generations and for a limited number of selected materials. This is because yield evaluation in early generations is very difficult and laborious and therefore had never been considered as a feasible option. In applying the directed evolution method to this study, we have succeeded in establishing effective and practical methods for doing this in the selection of higher yield mutants using gamma-ray irradiation, though only one cycle has been conducted so far. This is what distinguishes our method from conventional breeding methods.

The gamma-ray irradiation method used here is not “site-directed mutagenesis”; rather, it exactly matches with the two-step process of the directed evolution method. The social acceptance of gamma-ray irradiation is quite high and numerous useful practical mutant varieties have been developed in the past 60 years [29]. This may be because of the smaller number of mutations in the whole genome induced by gamma-ray [24], and the smaller number is an advantage in the development of practical varieties. The method proposed here has been proven to be useful in generating higher yield mutants, which hence contribute to the discovery of novel genes for higher yield. This is the main difference from CRISPR/Cas9 mutation.

### 3.4. Physiological Traits as the Underlying Causes behind Higher Yield Mutations

The culm length of FukuhibikiH8 was at most 6.6 cm longer than the original and this was the most obvious difference in this experiment. The culm and panicle length of YamadawaraH3 was also longer than its original variety, statistically. However, the differences were at most 3.2 cm for culm length and 0.6 cm for panicle length. Their culm length was around 80 cm and their panicle length was around 20 cm. The differences were too small to allow use of these traits as selection markers of higher yield mutant plants or lines. The phenotypic differences between the higher yield mutants and their original varieties were small and they were not always consistent over the years, even though they were statistically significant. The observed appearances were on the whole quite similar between the higher yield mutants and their originals, to the extent that the multi-replicated yield plots trials that we conducted had the appearance of a single cultivar cultivation (Figure 3). In addition to the above-mentioned results, the SNP patterns of higher yield mutants were the same as their original parents. The mutated points of the whole genome DNA sequences of YamadawaraH3 from Yamadawara and FukuhibikiH8 from Fukuhibiki were only 83 and 179, respectively [27]. The differences of their genome sequences from the original varieties were very limited.

We assumed that the causes of the higher yields of our selected mutants must have been physiological traits rather than phenotypical traits for the following three reasons. Firstly, the differences in traits between higher yield mutants and their originals were marginal and were not consistent over the years. Secondly, we did not use phenotypic traits for our selection of the mutants. Until now, high-yielding crop studies have been conducted by using phenotypic traits like semi-dwarf culms and grain or panicle sizes. The modifications of these traits were useful for upgrading yield stability or the improvement of sink capacity, but they could not directly increase crop productivity. The improvements obtained by such methods were different from the intrinsic yield increases obtained in this research. Thirdly, we observed lower leaf temperatures of FukuhikibiH8 compared with the original variety (Figure 4). The leaf temperature difference for FukuhibikiH8 was around 1.0 degree Celsius lower according to the measurements in the right photo, even though their phenotypical appearance was quite similar. MochidawaraH1 also showed a lower leaf temperature than its original in 2016 (Appendix A).

Our assumptions are as follows. Firstly, the higher yield mutants were selected as a result of their higher physiological productivities. Secondly, higher physiological productivities are expressed not only in terms of higher yield but also occasionally in terms of some other relating traits such as culm length, panicle length, seed set percentages, panicle sizes, grain sizes, or leaf temperature. The annotations of isolated higher yield gene candidates also suggest that the higher yield genes are physiological traits rather than phenotypical traits (data are not shown). Further functional studies including physiological traits such as leaf temperature and photosynthesis are necessary for the confirmation and identification of the higher yield genes of the mutants.

### 3.5. Some Other Considerations Concerning Our Methods

In this report, we employed pairwise replications for the yield trials of the mutants. Generally, crop yields are influenced by soil fertility conditions. In Appendix A, we could observe an undulating pattern in the yield over the various blocks in the same fields. This corresponded to the natural occurrence of differences in soil fertilities in the field. By using a pairwise replicated design, we could overcome the soil fertility differences which were considered to be the block variances of the ANOVA test (Appendix A).

In this experiment, we could successfully select higher yield mutants by individual plant evaluations. Even if we try to employ this method to study other major crops, like soybean, maize, or wheat, we expect that it would not be easy. The reasons are as follows. Firstly, it is easier to achieve uniform conditions in paddy fields, where water mediates the exchange of soluble nutrients in soil, than in upland fields. Secondly, rice is suitable for individual yield evaluation since a single plant only requires 0.045 m^2^ and produces around 40 g grain yield per plant (Appendix A). This enables us to evaluate many plants in a small area easily. Thirdly, rice is a self-pollinated crop that enables us to conduct easier genetic analysis and there are abundant available genomic information and tools to study rice.

## 4. Materials and Methods

### 4.1. Plant Materials and Irradiation Treatment

The dried seeds of five high-yielding rice varieties were irradiated with 250 Gy using ^60^Co source for 20 h at the Radiation Breeding Division, Institute of Crop Science, NARO (National Agriculture and Food Research Organization). The five original varieties are as follows: Fukuhibiki, the highest yield variety in Tohoku region; Oonari, the highest yield variety in Kanto region; Yamadawara, the highest yield variety for processing rice; Mochidawara, the highest yield variety of glutinous rice, and Akidawara, the highest yield variety for ordinary consumption. Three hundred grams of M_1_ seeds of each variety were used. Oonari M_1_ plants were grown in the summer of 2012 at the Yawara experimental field of NARO and the M_2_ seeds were harvested and stored in a refrigerator. The M_1_ seeds of the other four varieties were sown in rice seedling boxes and grown in a greenhouse in the winter season from November 2013 to April 2014.

### 4.2. Plant Selection from M_2_ Populations and Pedigree Selection of M_3_ to M_7_ Materials

In April 2014, the amount of M_2_ seeds of Fukuhibiki, Yamadawara, Mochidawara, and Akidawara harvested were 108 g, 218 g, 92 g, and 6 g, respectively. These seeds and the stored M_2_ seeds of Oonari were sown in seedling boxes and the 30-day seedlings of each M_2_ plant were manually transplanted in the paddy fields of the Radiation Breeding Division in May 2014 at plant intervals of 15 cm and row intervals of 30 cm. The numbers of individually transplanted M_2_ seedlings were around 2500, 4500, 4500, 2500, and 250 for Fukuhibiki, Oonari, Yamadawara, Mochidawara, and Akidawara, respectively. The numbers of harvested M_2_ plants were 1600, 3001, 3000, 2000, and 200, respectively. The total number of collected M_2_ plants was 9801. All the panicles of each plant were cut at the neck, collected into individual paper bags, and dried in a greenhouse for several days. The panicle weight of each plant was measured as the individual plant yield, and we selected M_2_ plants according to their yield. Apparently smaller and weaker plants were not included in the collection. In addition to this, border plants and plants adjacent to smaller plants or vacant spots were not collected since they had positional advantages over other plants. All the paper bags were numbered for the identification of the position of each plant in the fields so that the effect of non-uniform soil fertility could be eliminated.

According to the M_2_ plant yields, we selected 55, 155, 132, 106, and 16 M_2_ plants of Fukuhibiki, Oonari, Yamadawara, Mochidawara, and Akidawara, respectively. The total number of selected M_2_ plants and established M_3_ lines was 464. Ten M_3_ plants that originated from one selected M_2_ plant were transplanted in a row at plant intervals of 15 cm and row intervals of 30 cm in 2015. We planted the five original varieties as a comparison and each of them was followed by its 9 mutant lines. Seven plants in a line, excluding border plants, were harvested for yield evaluation in the same way as M_2_ plants. We observed and recorded heading dates, plant heights, leaf color, sterilities, and tiller formation of each line. Segregated lines were eliminated as candidates for the yield trials in the next step. In addition to this, we eliminated the neighboring lines of segregated ones. The neighboring lines had advantages because of the weak plants in the segregated lines. For example, a partially sterile plant can give an advantage to neighboring plants.

We selected a total of 12 higher yield mutant candidates according to the yield performances of M_3_ lines in 2016 (Table 1). In 2017, three mutants were selected among these 12 (Table 2). We added one more mutant FukuhibikiH8 in 2018 and 2019 (Table 3 and Table 4). This was a runner-up line in 2015 and was not included in the yield trials in 2016. However, because it showed good performance in 2017 with two replication yield trials (data not shown), we decided to include it, too. In 2019, the 5th block of FukuhibikiH8 with heavy fertilizer application was seriously damaged by panicle blast and we excluded the data of that block.

For establishing M_4_ lines from M_3_ lines, we selected the highest yield M_3_ plant in each line. In the M_5_, M_6_, and M_7_ generations, the pedigree selection and the maintenance of the lines were conducted in the same way.

### 4.3. Field Evaluations of Higher Yield Mutant Candidates in M_4_ to M_7_

The number of selected higher yield M_4_ mutant candidates was 2, 2, 2, 4, and 2 for Fukuhibiki, Oonari, Yamadawara, Mochidawara, and Akidawara, respectively, in 2016. We conducted two replications and two levels of fertilizer application in 2016, and the total number of plots for each mutant was four. Yield trials were conducted with the original varieties planted in plots adjacent to their corresponding mutants for the purpose of comparison. Three plants were transplanted in each hill manually. Each plot contained four rows and 132 hills. We excluded the borders of the plots and harvested 50 hills in each plot to measure the grain yield. Field observations were conducted for 3 traits at the seedling stage, 6 traits at the maximum tillering stage, and 19 traits at the maturing stage, according to the standard Japanese rice breeding scheme. Ten plants in each plot were picked for the measurement of culm length, panicle length, and the number of panicles before harvest. Five plants were picked for the measurement of seed set percentage and three panicles were measured for each plant. Air-dried whole plant weight, grain weight, brown rice weight, residual grain weight, and 1000 grain weight were measured after harvest. Brown rice weight was adjusted to water content by 15%. For data analysis, we employed the analysis of variance test (ANOVA).

Since yield advantages of higher yield candidates were not clear in the M_4_ yield trials, we conducted yield trails with 10-pairwise replications of M_5_ lines of Fukuhibiki, Oonari, Yamadawara, Mochidawara, and Akidawara, namely FukuhibikiH6, OonariH2, MochidawaraH1, YamadawaraH2, and YamadawaraH3, in 2017 (Appendix A). One of the ANOVA analyses is shown in Appendix A. The significant levels of 5% and 1% were calculated in the same way. The differences in the blocks are shown only in Appendix A. YamadawaraH2 and YamadawaraH3 were tested in paddy fields with standard fertilizer application. FukuhibikiH6, OonariH2, and MochidawaraH1 were tested in paddy fields with heavy fertilizer application. In 2018 and 2019, M_6_ and M_7_ YamadawaraH3 were tested with 8-pairwise replications in paddy fields with standard fertilizer application. In 2018 and 2019, M_6_ and M_7_ FukuhibikiH6, FukuhibikiH8, and MochidawaraH1 were tested with 8-pairwise replications in paddy fields with heavy fertilizer application. In addition to the above, in 2019, M_7_ FukuhibikiH6 and FukuhibikiH8 were tested in paddy fields with standard fertilizer application, with 8 and 6-pairwise replications, respectively. Since the number of replications was different for the two mutants in the standard fertilizer application field, Fukuhibiki was represented twice in comparison to each of them (Table 4). The methods of measurement were the same as the yield trail in 2016. Grain weight was adjusted to water content by 15%. The number of grains per panicle and the 1000 grain weight were not measured in 2018.

### 4.4. DNA Preparation and SNP Genotyping

To confirm the purity of the materials in this research, we conducted SNP genotyping analysis between higher yield mutants and their original varieties. For SNP genotyping, FukuhibikiH6, YamadawaraH3, MochidawaraH1, and their original irradiated varieties were used. In addition to these, different stocks of each of the original Fukuhibiki, Mochidawara, and Yamadawara were also used. Genomic DNA was prepared from fresh leaves using DNeasy Plant Maxi Kit (Qiagen Inc., Valencia, USA) and dissolved in 1× TE buffer (1 mM Tris-HCl, 0.1 mM EDTA, pH 8.0).

SNP genotypes were determined by using the 96.96 Dynamic Array IFC (96.96 IFC) chip according to the “SNP type 96 × 96 v1” protocol, except that the number of additional cycles after touchdown PCR was reduced from 34 to 30. Scanned data obtained with an EP1 reader (Fluidigm Inc.) were analyzed with SNP genotype analysis software (Fluidigm Inc.) and converted to scatter plot diagrams and allele data. SNP genotyping assays that gave clear and stable plot diagrams were selected to produce a set of 96 SNPs that were used in further experiments. SNP 96 markers are listed in Appendix A.

## Figures and Tables

**Figure 1 plants-09-01004-f001:**
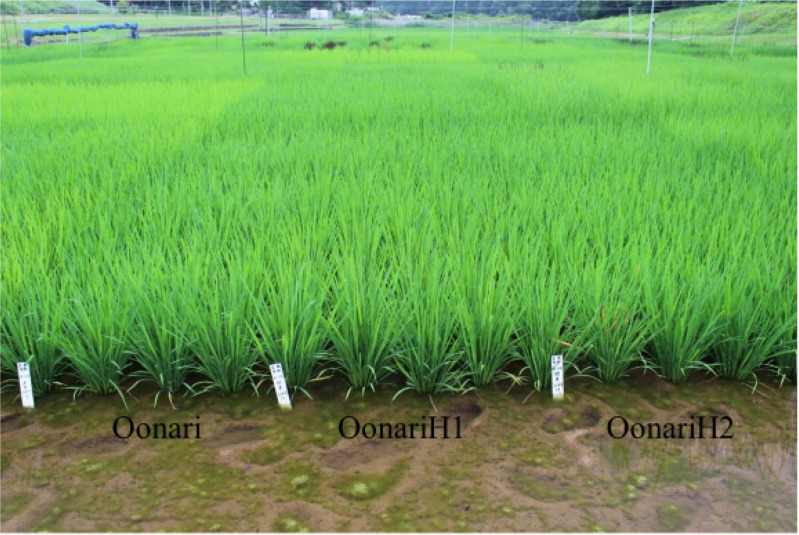
M_4_ OonariH1 and OonariH2 at maximum tillering stage.

**Figure 2 plants-09-01004-f002:**
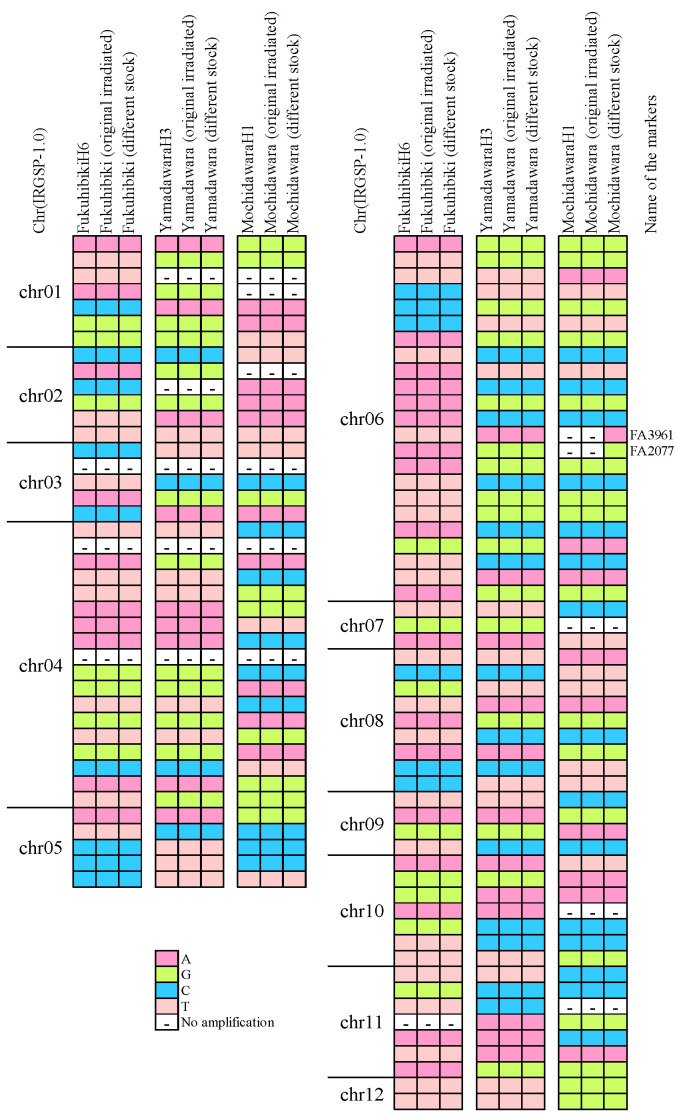
SNP genotyping of FukuhibikiH6, YamadawaraH3, and MochidawaraH1 by 96 SNP markers in 2017.

**Figure 3 plants-09-01004-f003:**
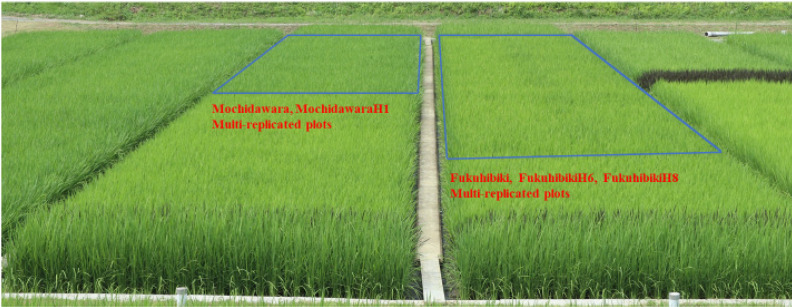
Multi-replicated yield trial plots of MochidawaraH1, FukuhibikiH6, and FukuhibikiH8 at maximum tillering stage in paddy fields in Ibaraki, Japan, in 2018.

**Figure 4 plants-09-01004-f004:**
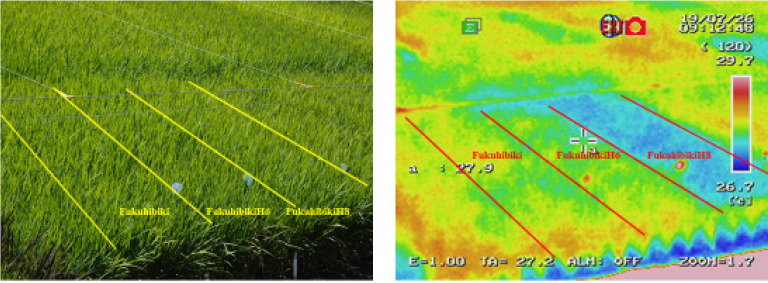
Yield evaluation (**left**) and thermo-camera observation (**right**) of FukuhibikiH6 and FukuhibikiH8 at maximum tillering stage in paddy field in Ibaraki, Japan, in 2019. Note: thermo-camera was “InfReC R300SR-S” of AVIONICS CO., LTD.

**Table 1 plants-09-01004-t001:** Yield trial of mutants in paddy field with standard fertilizer application in Ibaraki, Japan in 2016.

Name of the Lines or Varieties	Gen.	Heading Date	Maturity Date	Culm Length (cm)	Panicle Length (cm)	No. Panicles (/m^2^)	Lodging	Air-Dried		Ratio (%)	Brown Rice w. (t/ha)	Ratio (%)	Residual Rice (%)	Seed Set (%)	1000 b.r.w. (g)
								Whole w. (t/ha)	Grain w. (t/ha)						
FukuhibikiH6	M_4_	Jul.27	Sept.5	64.2	18.0	306	0.0	16.0	7.07	102.6	5.92	102.6	0.2	89.4	24.0
FukuhibikiH7	M_4_	Jul.26	Sept.1	67.5	18.6	296	0.0	15.8	6.73	97.7	5.68	98.4	0.1	88.8	23.9
Fukuhibiki		Jul.27	Sept.1	64.7	18.7	300	0.0	16.4	6.89	100.0	5.77	100.0	0.2	86.6	23.4
OonariH1	M_4_	Aug.9	Sept.30	72.4	27.1	268	0.5	17.1	9.62	99.3	8.15	101.4	0.1	81.9	22.2
OonariH2	M_4_	Aug.8	Sept.28	70.5	27.1	268	0.5	16.8	9.82	101.3	8.15	101.4	0.1	85.4	21.1
Oonari		Aug.8	Sept.30	74.0	26.2	287	0.5	17.1	9.69	100.0	8.04	100.0	0.1	80.9	21.7
YamadawaraH1	M_4_	Aug.9	Sept.30	78.0	20.6	288	0.0	17.4	8.43	101.4	7.18	102.0	0.2	86.7	22.7
YamadawaraH2	M_4_	Aug.9	Sept.30	81.1	21.1	282	0.5	17.6	8.71	104.8	7.38	104.8	0.1	85.8	22.7
YamadawaraH3	M_4_	Aug.8	Sept.30	81.5	21.2	297	1.0	18.2	8.93	107.5	7.52	106.8	0.2	86.5	23.4
YamadawaraH4	M_4_	Aug.10	Sept.30	77.5	20.9	286	0.0	17.4	8.62	103.7	7.29	103.6	0.2	85.5	22.7
Yamadawara		Aug.9	Sept.30	78.3	20.8	293	0.0	16.8	8.31	100.0	7.04	100.0	0.2	85.2	22.8
MochidawaraH1	M_4_	Aug.12	Oct.6	84.5	26.5	208	1.0	18.1	9.91	104.4	8.16	103.3	0.1	87.7	22.6
MochidawaraH2	M_4_	Aug.13	Oct.6	86.9	26.4	196	1.0	17.8	9.51	100.2	7.89	99.9	0.1	84.0	22.3
Mochidawara		Aug.12	Oct.6	84.5	26.2	209	1.0	17.7	9.49	100.0	7.90	100.0	0.2	87.2	22.4
AkidawaraH1	M_4_	Aug.12	Sept.27	77.2	20.9	291	0.5	17.7	7.96	97.9	6.78	98.1	0.1	92.4	21.0
AkidawaraH2	M_4_	Aug.12	Sept.27	80.7	22.3	290	0.5	17.8	7.80	95.9	6.68	96.7	0.1	90.8	21.4
Akidawara		Aug.13	Sept.27	76.5	21.1	291	0.5	18.6	8.13	100.0	6.91	100.0	0.1	89.3	21.7

Lodging: 0 means none and 9 means whole lodged. Gen.: generation, w.: weight, b.r.w.: brown rice weight. Brown rice weight was adjusted to water content by 15%. Fertilizer application was 70 kg nitrogen/ha.

**Table 2 plants-09-01004-t002:** Yield trial of mutants in paddy fields in Ibaraki, Japan in 2017.

Mutant (upper) Original v. (lower)	Gen.	Heading Date	Maturity Date	Culm Length (cm)	Panicle Length (cm)	No. Panicles (/m^2^)	Lodging	Air-Dried		Ratio (%)	Seed Set (%)	No. Grains/Panicle	1000 Grain w. (g)
								Whole w. (t/ha)	Grain w. (t/ha)				
FukuhibikiH6	M_5_	24-Jul *	9-Sep	70.5	21.8 **	247	0.0	13.77	7.98 *	105.1	94.2	106.2	28.8 *
Fukuhibiki		25-Jul	9-Sep	70.2	21.3	246	0.0	13.40	7.59	100.0	93.4	105.5	28.1
YamadawaraH3	M_5_	16-Aug	1-Oct	84.0 *	20.3 *	289	0.0	19.42	9.71 **	103.7	90.5	133.9	27.6
Yamadawara		16-Aug	1-Oct	82.4	20.0	289	0.0	19.64	9.36	100.0	91.2	131.8	27.2
MochidawaraH1	M_5_	23-Aug	18-Oct	91.8	26.1	228	0.0	29.51	10.40 **	104.8	82.5 **	172.4	28.3
Mochidawara		23-Aug	18-Oct	92.0	26.0	221	0.0	29.18	9.92	100.0	80.1	167.2	28.4

Original v.: Original variety, Gen.: Generaton, w.: weight. Lodging: 0 means none and 9 means whole lodged. * and ** mean 5% and 1% signigicant levels, respectively. Grain weight was adjusted to water content by 15%. YamadawaraH3 yield trials were cultivated in standard fertilizer application which was 100 kg nitrogen/ha. FukuhibikiH6 and MochidawaraH1 were cultivated in heavy fertilizer application which was 200 kg nitrogen/ha.

**Table 3 plants-09-01004-t003:** Yield trial of mutants in paddy fields in Ibaraki, Japan in 2018.

Mutant (Upper) Original v. (Lower)	Gen.	Heading Date	Maturity Date	Culm Length (cm)	Panicle Length (cm)	No. Panicles (/m^2^)	Lodging	Air-Dried		Ratio (%)	Seed Set (%)
								Whole w. (t/ha)	Grain w. (t/ha)		
FukuhibikiH6	M_6_	28-Jul	8-Sep	84.6 **	20.4	346	0.9	18.28	9.89 *	101.2	91.3
FukuhibikiH8	M_6_	28-Jul	8-Sep	87.0 **	21.0	354	1.8	18.64 **	10.18 **	104.1	92.1
Fukuhibiki		28-Jul	8-Sep	82.6	20.5	365	0.9	18.19	9.77	100.0	92.4
YamadawaraH3	M_6_	10-Aug **	2-Oct	83.3	21.4 **	343	0.0	21.53	10.18 **	102.7	84.9
Yamadawara		13-Aug	2-Oct	81.3	21.0	322	0.0	21.25	9.92	100.0	84.5
MochidawaraH1	M_6_	15-Aug	6-Oct	91.4	26.5	254 **	2.6	24.81	11.01	100.5	80.8
Mochidawara		15-Aug	6-Oct	91.5	26.2	266	2.4	25.19	10.95	100.0	78.2

Original v.: Original variety, Gen.: Generaton, w.: weight. Lodging: 0 means none and 9 means whole lodged. * and ** mean 5% and 1% signigicant levels, respectively. Grain weight was adjusted to water content by 15%. YamadawaraH3 yield trials were cultivated in standard fertilizer application which was 100 kg nitrogen/ha. FukuhibikiH6, FukuhibikiH8, and MochidawaraH1 were cultivated in heavy fertilizer application which was 200 kg nitrogen/ha.

**Table 4 plants-09-01004-t004:** Yield trial of mutants in paddy fields in Ibaraki, Japan in 2019.

Mutant (Upper) Original v. (Lower)	Gen.	Heading Date	Maturity Date	Culm Length (cm)	Panicle Length (cm)	No. Panicles (/m^2^)	Lodging	Air-Dried		Ratio (%)	Seed Set (%)	No. Grains/Panicle	1000 Grain w. (g)
								Whole w. (t/ha)	Grain w. (t/ha)				
	(Standard fertilizer applicaton with 100 kg nitrogen/ha)												
FukuhibikiH6	M_7_	29-Jul *	5-Sep	70.0 *	19.4 *	318	1.0 **	14.42	6.25 **	108.1	89.1	71.9	27.4 *
Fukuhibiki		28-Jul	5-Sep	68.3	18.5	312	0.0	13.97	5.78	100.0	87.5	69.5	26.7
FukuhibikiH8	M_7_	29-Jul *	5-Sep	73.2 **	18.6	306	0.6 *	14.96	7.23 **	122.5	91.2 *	82.9 **	28.5 **
Fukuhibiki		28-Jul	5-Sep	68.5	18.5	310	0.0	14.15	5.90	100.0	87.0	71.2	26.7
YamadawaraH3	M_7_	7-Aug	20-Sep	81.8 **	20.9 *	322	0.3 *	16.61	8.49	103.9	90.3	91.8	28.8 **
Yamadawara		8-Aug	20-Sep	78.6	20.3	323	0.0	16.17	8.17	100.0	90.3	92.8	27.3
	(Heavy fertilizer application with 200 kg nitrogen/ha)												
FukuhibikiH6	M_7_	31-Jul	5-Sep	82.3 **	19.2 *	404	1.9 **	17.70	8.65 **	109.3	93.0 **	88.9 **	24.0
FukuhibikiH8	M_7_	31-Jul	5-Sep	84.6 **	20.1	412	2.3 **	17.70	9.17 **	115.8	94.2 **	92.3 **	24.3
Fukuhibiki		30-Jul	5-Sep	78.0	19.8	428	1.1	17.40	7.91	100.0	85.8	76.1	24.2
MochidawaraH1	M_7_	10-Aug	2-Oct	95.3	27.3 *	307	8.4	22.31	10.94	101.5	91.2 *	130.1	27.3
Mochidawara		10-Aug	2-Oct	94.6	28.0	294	8.6	21.73	10.78	100.0	88.7	133.3	27.6

Original v.: Original variety, Gen.: Generation, w.: weight. Lodging: 0 means none and 9 means whole lodged. * and ** mean 5% and 1% signigicant levels, respectively. Grain weight was adjusted to water content by 15%.

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
