# Peer review of "The Selection of Gamma-Ray Irradiated Higher Yield Rice Mutants by Directed Evolution Method"

_plants, 2020, doi:10.3390/plants9081004_

Round 1
Reviewer 1 Report
Physical Mutagenesis has been widely used for generating mutants for functional genomics and plant breeding. In this manuscript, gamma-ray irradiation was used to treat rice varieties to develop higher yield rice lines. A total of five high-yielding rice varieties were selected for irradiation. The authors generated 9,801 irradiated M2 plants, from which 464 individuals showed higher panicle weight. Finally, they selected 4 higher yield mutants with 1.2 to 22.5% higher yield than their original parental lines. Genotyping with SNP markers showed that Only one difference was observed from the irradiated Mochidawara and there was no difference between the remaining three higher yield mutants and their irradiated original varieties. The authors then suggested that the higher yields of the selected mutants were likely to have been caused by physiological traits rather than phenotypical traits. Generally, the authors carried out lots of works and the developed lines showed the high yield potential.
As only limited SNP markers were used for genotyping, it is reasonable for the authors to identify no difference between mutated and parental lines. Authors may carry out RNA-Seq or genome sequencing analysis to identify genetic variation between parental and irradiated lines.
Detailed description should be provided on how to carry out statistical analysis.
Figure 2 can be cited as a Supplementary figure.
Based on the converted pdf file, Figure 3 is not clearly recognized and higher resolution is required
Author Response
Reviwer1
Thank you for your constructive comments. I have followed all of your comments. I have added and modified by using red letters in the attached files: plants-886267(1)-KATO. Following is the reply to your comments.
Comment: As only limited SNP markers were used for genotyping, it is reasonable for the authors to identify no difference between mutated and parental lines. Authors may carry out RNA-Seq or genome sequencing analysis to identify genetic variation between parental and irradiated lines.
Reply: We have added the information of genome sequence analysis in Discussion L279.
L279: The mutated points of whole genome DNA sequences of YamadawaraH3 from Yamadawara and FukuhibikiH8 from Fukuhibiki were only 83 and 179, respectively [27]. The differences of their genome sequences from the original varieties were very limited.
Comment: Detailed description should be provided on how to carry out statistical analysis.
Reply: In accordance with reviewer’s suggestion, we have added Table S4 which showed the process of statistical analysis for the extracted data from Table S3. We have also added relating explanations in Results L128 and Materials and Methods L386.
L128: Table S4 shows the statistical procedure of grain yield of FukuhibikiH6 and Fukuhibiki in 2017. Other statistical significances were calculated in the same way.
L386: One of the ANOVA analysis was shown in Table S4. The significant levels of 5% and 1% were calculated in the same way. The difference of the blocks was shown only in Table S4.
Comment: Figure 2 can be cited as a Supplementary figure.
Reply: We have moved it to supplementary figure.
Comment: Based on the converted pdf file, Figure 3 is not clearly recognized, and higher resolution is required
Reply: The resolution of original attached file is high, we are going to consult with editing section about it.

Reviewer 2 Report
This manuscript identified several high yielding mutant lines using gamma-ray in rice. Based on the year by year yield test, some good mutant lines were developed. This result suggested the new approach for mutation breeding to improve yield. It can be acceptable with some minor changes.
- In 2018-2019 yield tests, FukuhibikiH8 was used. However, there was no any discription for FukuhibikiH8 in M4-M5 generations. Why added the FukuhibikiH8 late?
- In Table 4, control cultivar Fukuhibiki was used in twice. Is there any reason?
- The high-yielding mutant lines were selected based on phenotype data and yield testing. This method was called as directed evolution method in this study. However, this method was normal procedure in mutation breeding. Need more description for directed evolution method.
Author Response
Reviewer2
Thank you for your constructive comments. I have followed all of your comments. I have added and modified by using red letters in the attached files: plants-886267(1)-KATO. Following is the reply to your comments.
Comment: In 2018-2019 yield tests, FukuhibikiH8 was used. However, there was no any discription for FukuhibikiH8 in M4-M5 generations. Why added the FukuhibikiH8 late?
Reply: We have added the information on FukuhibikiH8 at L360. We have also divided the paragraph.
L360: It was a runner-up line in 2015 and was not included in the yield trials in 2016. However, because it showed good performances in 2017 with two replication yield trials (data not shown), we decided to include it, too.
Comment: In Table 4, control cultivar Fukuhibiki was used in twice. Is there any reason?
Reply: We have added the explanation about it at L395.
L395: Because the number of replications were different for the two mutants in standard fertilizer application filed, Fukuhibiki was represented twice in comparison to each of them (Table 4).
Comment: The high-yielding mutant lines were selected based on phenotype data and yield testing. This method was called as directed evolution method in this study. However, this method was normal procedure in mutation breeding. Need more description for directed evolution method.
Reply: We have added the following in Discussion L245 and L251 for clarifying the difference between this method and conventional methods. We have also divided the paragraph.
L245: like higher activity
L251: Conventionally, breeders conduct the yield trials only in advanced generations and for a limited number of selected materials. This is because yield evaluation in early generations is very difficult and laborious and therefore had never been considered as a feasible option. In applying the directed evolution method to this study, we have succeeded in establishing effective and practical methods for doing this in the selection of higher yield mutants using gamma-ray irradiation, though only one cycle has been done so far. This is what distinguishes our method from conventional breeding methods.

Round 2
Reviewer 1 Report
The authors have taken care of the comments.